# Gaussian process emulation for exploring complex infectious disease models

**Anna M. Langmüller**[1,2,3*], **Kiran A. Chandrasekher**[1], **Benjamin C. Haller**[1],
**Samuel E. Champer**[1], **Courtney C. Murdock**[4,5,6], **Philipp W. Messer**[1*]

**1** Department of Computational Biology, Cornell University, Ithaca, New York, United States of America,
**2** Department of Mathematics, University of Vienna, Vienna, Austria, **3** Aarhus Institute of Advanced
Studies, Aarhus University, Aarhus, Denmark, **4** Department of Entomology, Cornell University, Ithaca,
New York, United States of America, **5** Cornell Institute of Host-Microbe Interactions and Disease, Cornell
University, Ithaca, New York, United States of America, **6** Center for the Ecology of Infectious Diseases,
University of Georgia, Athens, Georgia, United States of America

* annamaria.langmueller@aias.au.dk, annamaria.langmueller@gmail.com (AML); messer@cornell.edu
(PWM)

Melbourne Faculty of Science, AUSTRALIA

**Peer Review History:** PLOS recognizes the
benefits of transparency in the peer review
process; therefore, we enable the publication
of all of the content of peer review and
author responses alongside final, published
articles. The editorial history of this article is
available here: https://doi.org/10.1371/journal.
pcbi.1013849

## Abstract

Epidemiological models that aim for a high degree of biological realism by simulating
every individual in a population are unavoidably complex, with many free parameters,
which makes systematic explorations of their dynamics computationally challenging.
In this study, we demonstrate how Gaussian Process emulation can overcome this
challenge. To simulate disease dynamics, we developed an abstract individual-based
model that is loosely inspired by dengue, incorporating some key features shaping
dengue epidemics such as social structure, human movement, and seasonality. We
focused on three epidemiological metrics derived from the individual-based model
outcomes — outbreak probability, maximum incidence, and epidemic duration — and
trained three Gaussian Process surrogate models to approximate these metrics. The
GP surrogate models enabled the rapid prediction of these epidemiological metrics
at any point in the eight-dimensional parameter space of the original model. Our
analysis revealed that average infectivity and average human mobility are key drivers
of these epidemiological metrics, while the seasonal timing of the first infection can
influence the course of the epidemic outbreak. We used a dataset comprising more
than 1,000 dengue epidemics observed over 12 years in Colombia to calibrate our
Gaussian Process model and evaluated its predictive power. The calibrated Gaussian
Process model identified a subset of municipalities with consistently higher average
infectivity estimates; the notable overlap between these municipalities and previously
reported dengue disease clusters suggests that statistical emulation can facilitate
empirical data analysis. Overall, this work underscores the potential of Gaussian
Process emulation to enable the use of more complex individual-based models in
epidemiology, allowing a higher degree of realism and accuracy that should increase
our ability to control diseases of public health concern.

**Data availability statement:** The source code for the individual-based disease transmission model implemented in C++ is available on GitHub at https://github.com/AnnaMariaL/DengueSim. Simulated data, pre-trained Gaussian process models, Jupyter notebooks demonstrating their use, and all data and code required to reproduce the results and figures presented in this study are available on GitHub at https://github.com/AnnaMariaL/DengueSim-GP.

**Funding:** This project & AML have received funding from the European Union's Horizon2020 (https://research-and-innovation.ec.europa.eu/funding/funding-opportunities/funding-programmes-and-open-calls/horizon-2020_en) research and innovation program under the Marie Sklodowska-Curie grant agreement No. 101025586. PWM was supported by the National Institutes of Health (https://grants.nih.gov/funding/activity-codes/r35) under award R35GM152242. The funders had no role in study design, data collection and analysis, decision to publish, or preparation of the manuscript.

**Competing interests:** The authors have declared that no competing interests exist.

## Author summary

Detailed individual-based models can capture a high degree of realism, but their complexity often makes them too slow or cumbersome to explore fully. In our work, we explore how Gaussian Process emulation — a statistical method for building fast, accurate surrogate models — can help overcome this challenge. First, we developed an individual-based model that simulates disease spread in a population, accounting for features such as social structure, human mobility, and seasonal variation in infection risk. We then trained a Gaussian Process surrogate model on epidemiological metrics derived from the outputs of this individual-based model, which allowed us to predict these metrics almost instantly across a wide range of parameter values. This approach made it possible to systematically explore which factors drive simulated epidemics. We found that two variables — average infectivity and average mobility — had the greatest influence on whether and how outbreaks occurred. Our results demonstrate that Gaussian Process emulation offers a practical and powerful way to study complex disease systems. While we applied this approach to infectious disease transmission, the underlying method can be useful for analyzing many other types of detailed, simulation-based models.

## Introduction

Simulation models that describe individual organisms — often referred to as individual-based or agent-based models — have become well-established research tools across numerous scientific disciplines [1]. In the field of epidemiology, such models have provided valuable insights into the dynamics of pathogen and disease spread and have facilitated rigorous evaluation of planned intervention strategies, making them an integral part of modern epidemiological research [2–6]. Recent computational advances, combined with the development of comprehensive simulation frameworks [2,7], have enabled the creation of epidemiological models with unprecedented realism, including details such as fine-scale human movement [5,8] and specific larval breeding sites for mosquito vectors [3].

While enhanced biological realism in simulation models has undeniably deepened our understanding of epidemiological processes, it also introduces increased complexity because of the level of detail being simulated, which comes at a substantial computational cost. As simulations become more realistic, they also become more parameter-rich, making it increasingly difficult to identify the key drivers of disease dynamics. This is partly because parameters often interact in complex, non-linear ways, complicating efforts to quantify the contribution of any single factor to model outcomes. Global sensitivity analysis can help by quantifying the relative contribution of each parameter — as well as their interactions — to model outcomes. For example, the Sobol method [9] is a global sensitivity analysis approach that is able to assess complex, non-linear parameter interactions by partitioning the observed

variance in model outcomes into relative contributions from single parameters as well as interactions between two or more parameters. This allows researchers to gain a deeper understanding of the key drivers of the disease dynamics observed in a simulation.

Global sensitivity analysis can provide valuable insights into model dynamics, but generating sufficient data for robust sensitivity analysis can be computationally demanding. For simulations that explicitly model each individual, computational demand typically scales at least linearly with population size and in the worst case can increase quadratically when intricate behaviors or pairwise interactions are modeled. Unlike mathematical models based on ordinary differential equations [10], which can often be examined analytically or with relatively modest computational effort, such simulation models often require thousands of runs to fully explore their complex parameter spaces. This makes comprehensive parameter space exploration particularly challenging for high-dimensional models, even when they are optimized for runtime performance, leading to the well-known "Curse of Dimensionality" [11].

Statistical emulation [11] can address this problem by constructing a fast, predictive surrogate model based on a limited set of simulation runs. Once trained, the emulator can predict the simulation model's outputs across the full parameter space in a fraction of the time it would take to generate the same outputs with the original simulation model. This allows efficient execution of tasks such as sensitivity analysis, model output exploration, and model calibration, at a resolution that would be infeasible using the simulation model directly. In essence, statistical emulation facilitates the extraction of meaningful insights from a complex simulation model by substantially reducing the computational costs associated with its analysis. A well-designed emulator can reduce computational runtimes from days to mere seconds, dramatically expanding the scope of analysis that is within practical reach. While emulators often rely on statistical methods, recent advances increasingly incorporate machine learning techniques [12] such as random forests, neural networks, and Gaussian Processes [12–16].

Gaussian Processes (GPs), first introduced in the 1960s within the field of geostatistics [17,18], are among the most widely used statistical emulators [13,14] and have been successfully applied across diverse disciplines [19–21]. GPs are non-parametric models that define a distribution over functions based on observed data. A key advantage of GPs over other machine learning techniques, such as conventional support vector machines or neural networks, lies in their Bayesian foundations, which allow GPs to provide confidence intervals alongside their predictions. This uncertainty quantification enables efficient sampling of additional training data from regions with the greatest uncertainty, facilitating active learning [22] that can quickly produce highly accurate emulators. Furthermore, with the availability of advanced software packages supporting GPU acceleration, the computational efficiency of GPs has improved at an astonishing pace in recent years, making them an increasingly attractive research tool [23].

In epidemiology, the ability of GPs to efficiently extrapolate between sparse data points is often utilized for estimating disease incidence counts in areas where data is missing or unobserved [24,25]. GPs also serve as valuable forecasting tools [21,26] and are key components of early-warning systems [27]. Furthermore, as emulators of complex, computationally intensive simulation models, GPs facilitate the calibration of these models to empirical data by helping to adjust parameter values in order to fit the simulation model to real-world data [28–30]. Notable examples of using GPs as emulators to better understand complex simulation models include recent studies that applied GP emulation to the OpenMalaria model (an advanced simulation model developed to simulate malaria transmission and control [30]) to explore key drivers of the spread of drug-resistant *Plasmodium falciparum* [31], and to assess the effectiveness of various malaria intervention strategies [32,33]. These applications, which often incorporate variance-based sensitivity analyses, demonstrate the power of GP emulation, but are typically highly tailored to a specific context and require a deep familiarity with a particular disease system.

In this study, we build upon prior applications of Gaussian Process (GP) emulation for sensitivity analysis in epidemiological contexts (e.g., [32,33]) by constructing a general framework that illustrates their applicability and practical benefits. Specifically, we developed an abstract, generalizable simulation model designed to showcase how GP emulation can

efficiently support variance-based sensitivity analysis across high-dimensional parameter spaces. This work illustrates the broader methodological potential of GP emulation for accelerating and interpreting complex epidemiological simulations. Our disease transmission model, which is loosely inspired by dengue, simulates the progression of an epidemic through a population of explicitly simulated individual humans, and incorporates some key features that shape dengue epidemics such as social structure, human movement, and seasonality. Dengue poses a growing global health threat [34], with cases rapidly increasing due to urbanization [35] and climate change that has expanded the habitat of *Aedes* mosquitoes, the primary vectors of the dengue virus [36,37]. Using epidemiological metrics derived from outcomes from this simulation model, we trained Gaussian Process surrogate models to predict outbreak probability, maximum incidence, and outbreak duration with high efficiency, enabling a comprehensive analysis of the system's behavior across its broad parameter space.

## Methods

### Individual-based model

We implemented an individual-based model (IBM; in epidemiology, the terms "individual-based" and "agent-based" are used largely interchangeably to describe models that simulate individual entities and their interactions [1]; following this convention, we will use "individual-based model" as a general term for such modeling approaches) in C++ that simulates and tracks disease transmission and includes several parameters related to infection probability, human movement, and social structure — three key features shaping dengue epidemics [38–40].

In our model, each individual is explicitly simulated, and each individual follows probabilistic and structural rules regarding infection and movement (though they lack adaptive, anticipatory, or learning behaviors). The model is designed not to replicate a specific empirical system, but to illustrate the relative importance of parameters and their interactions in influencing the course of simulated epidemic outbreaks. A detailed IBM description can be found in S1 Text. All model parameters are summarized in Table 1.

Each simulation begins by generating 10,000 locations that are each home to a group of susceptible individuals. The number of individuals per location is sampled from a negative binomial distribution fitted to the demography of Iquitos, Peru — a well-studied dengue transmission hotspot [38,41]. These locations are then randomly organized into non-overlapping family clusters, with the "family cluster size" parameter controlling the number of locations per cluster. Social structure, controlled by the "social structure" parameter, influences the likelihood that individuals interact within their family cluster. Human movement — the number of visits to locations per day (in addition to the family home) — is sampled from

Table 1. Parameters of the individual-based disease transmission model.

| Parameter | Description | Default | Range |
|---|---|---|---|
| Average infectivity | The average infection probability across a year, which removes the effect of seasonality | 0.015 | [0, 0.03] |
| Seasonality strength | A scaling factor between 0 and 1 that controls the magnitude of seasonal variation in infection probability | 0.5 | [0, 1] |
| First case timing | Determines the timing of the first case relative to the seasonal peak in infection probability | 0 | [0, 1] |
| Infectious period | The average number of days an individual remains infectious, with actual days determined by probabilistically rounding to the nearest integers around the specified value | 5 | [4, 6] |
| Average mobility | The average number of visits a human makes to locations per day, in addition to their family home | 2 | [1, 5] |
| Mobility skewness | The success probability in the negative binomial distribution that determines the number of visits a human makes per day; a lower value results in greater variance in the daily visit count | 0.5 | [0.05, 0.95] |
| Social structure | The probability of a visit occurring within the family cluster of the individual moving | 0.5 | [0, 1] |
| Family cluster size | The average number of locations assigned to each cluster, with actual sizes determined by probabilistically rounding to the nearest integers around the specified value | 5 | [1, 20] |

a negative binomial distribution, defined by the "average mobility" and "mobility skewness" parameters. The social structure of the model is depicted in S1 Fig.

The disease is introduced by infecting a single randomly chosen individual. Infected individuals remain contagious for a number of days specified by the "infectious period" parameter, after which they recover and gain lasting immunity (and thus cannot become reinfected). When a susceptible individual visits a location that was visited by infectious individuals the day before, the likelihood of infection from each previous infectious visitor is determined by the infection probability. In the context of dengue, seasonal fluctuations in this infection probability can be interpreted as reflecting changes in mosquito abundance over the year. Although dengue is transmitted by mosquitoes, we do not model individual mosquitoes in our abstract simulation framework. This choice was driven by the very limited dispersal ability of *Aedes aegypti* [42], the primary vector for dengue in the Americas, which predominantly bites during daylight hours [43]. Consequently, human movement patterns tend to be more influential than mosquito movement in shaping dengue dynamics [38,40,44].

The infection probability is defined by a cosine function with three parameters: (i) the "average infectivity" parameter ($\alpha_0$), representing the average infection probability over the course of a year (365 days); (ii) the "seasonality strength" parameter ($\alpha_{season}$), controlling the magnitude of seasonal variation in infection probability; and (iii) the "first case timing" parameter ($t_{first}$), defining the horizontal shift of the cosine function and thus the timing of the first case relative to the peak infection probability due to seasonality. Together, these parameters define the infection probability at any given day $t$ in the year:

$$p_{infection}(t) \; = \; \alpha_0 \; * \; (1 \; + \; \alpha_{season} \; * \; cos(2\pi \; * \; (t/365 - t_{first})))$$

The IBM progresses by daily timesteps and continues until there are no infectious individuals left. The output consists of daily counts of individuals in each infection state (susceptible, exposed, infectious, and recovered). For each combination of parameters, we used 100 replicate simulation runs to calculate three metrics of the simulated epidemics: (i) outbreak probability, defined as the proportion of simulation runs in which more than 0.1% of the population becomes infected; (ii) maximum disease incidence ($i_{max}$), defined as the highest proportion of infectious individuals seen in any timestep; and (iii) outbreak duration, defined as the timespan in days from the first infectious case to the recovery of the last infectious individual. Because $i_{max}$ and outbreak duration are only meaningful when an outbreak occurs, these metrics were calculated conditional on outbreak occurrence; we conducted additional simulations as needed to obtain 100 such runs for each parameter combination.

We systematically varied the eight parameters outlined above to explore how the simulated epidemics change across the parameter space. Across the full range of parameters (Table 1), the three metrics vary significantly: the average outbreak probability is 0.79, ranging from 0 to 1; the average $i_{max}$ is 0.67, ranging from 0.0003 to 0.99; and the average duration is 63.88 days, ranging from 19.65 to 424.15 days.

## Gaussian processes

We trained Gaussian Process (GP) surrogate models on input-output pairs from the IBM to efficiently approximate its behavior [12] (Fig 1). We implemented GPs in Python (v3.10.6) using the GPyTorch library (v1.11) [23] for efficient GP modeling, and the torch.cuda module from the PyTorch package (v2.0.1) [45] for GPU acceleration with NVIDIA GPUs. The implementation was inspired by a GP surrogate model previously used to study the efficiency of gene drives in rat populations [20].

We trained a separate GP model for each of the three outbreak metrics described above: outbreak probability, maximum incidence $i_{max}$, and epidemic duration. While the IBM outputs for outbreak probability and $i_{max}$ are bounded between 0 and 1, epidemic duration spans a much wider range (initial training dataset (N = 5,000): 19.65 — 424.15 days). To manage the variance in epidemic duration and improve the GP's ability to predict longer epidemics, we applied a logarithmic transformation to the outbreak duration.

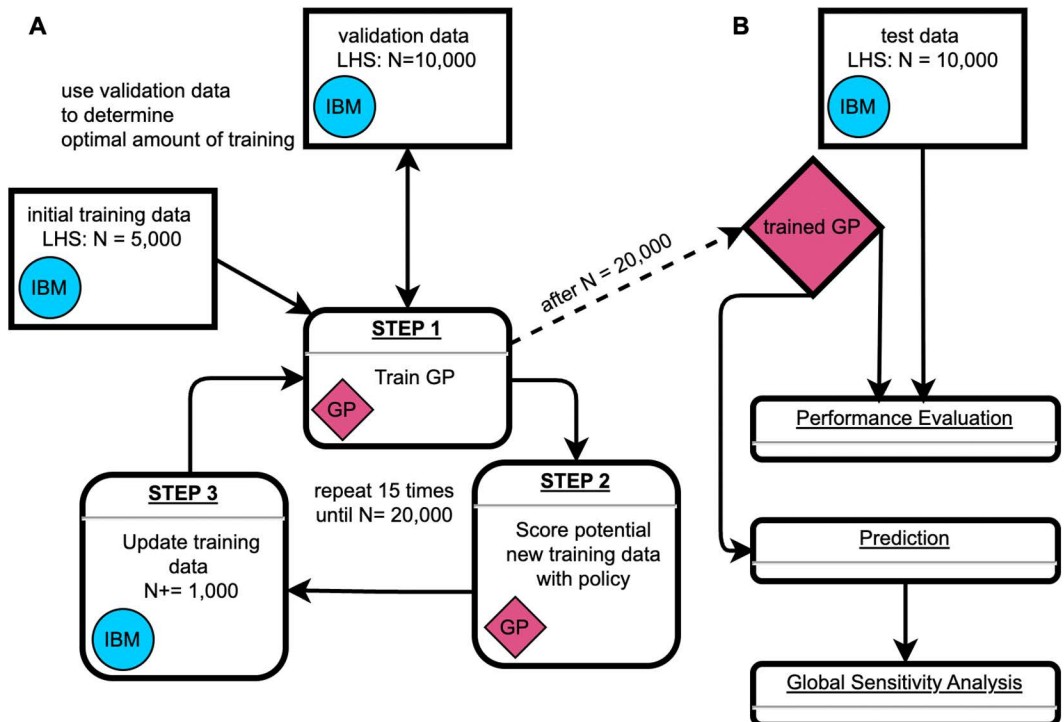

**Fig 1. Gaussian Process training & emulation workflow.** (A) Gaussian Process (GP) training loop [22]. The GP training begins with an initial training dataset consisting of a Latin hypercube sample (LHS) of 5,000 data points generated from the input domain (Table 1) using the individual-based simulation model (IBM). During training, the GP is evaluated against a validation dataset of 10,000 data points to determine the optimal amount of training iterations and prevent overfitting. After each training cycle, $10^7$ potential new data points are scored based on a policy that considers their predicted value and 95% confidence interval. In each iteration of the training loop, 1,000 additional data points are sampled from these $10^7$ candidate points, with sampling probability proportional to their policy scores. The newly selected data points are then simulated using the IBM, added to the training dataset, and the next training round begins. (B) Use of the trained GP. After training, the GP is tested using an independent dataset of 10,000 LHS data points to evaluate its performance. The trained GP can then be used for rapid predictions, enabling large-scale global sensitivity analyses.

The covariance function — or kernel — of a GP determines how much the response values of different input points covary [14]. Thus, the choice of kernel is crucial in shaping the GP's predictions. We selected a Matérn kernel with $v = 0.5$, which corresponds to the exponential kernel. This kernel is capable of capturing abrupt changes in function values [22]. We applied the same kernel type across all three GPs.

**Gaussian process training loop.** We implemented a three-step active learning loop (Fig 1A) in which the GP iteratively identified regions of high predictive uncertainty, selected new training points in those regions, and retrained on the expanded dataset [22].

**Step 1: GP training.** We trained each GP for 16 rounds. The first round used a Latin hypercube sample (LHS) of 5,000 points. Latin hypercube sampling is a stratified sampling method that efficiently covers the entire input domain (Table 1). The remaining 15 rounds used active training (Fig 1A). For GP training, we utilized the Adam optimizer from PyTorch [45] with a learning rate of 0.01. In each training round, we trained the GP for 30,000 iterations, with a model snapshot saved every 1,000 iterations. To avoid overfitting, we evaluated all 30 snapshots against a separate validation dataset consisting of 10,000 LHS points. We selected the snapshot with the lowest root mean square error (RMSE) on the validation dataset for step 2 in the training loop.

**Step 2: Data scoring.** In each active learning round, we scored $10^7$ LHS points using two distinct policies [20]. These scores are used as probability weights to select 1,000 new data points to expand the training data. Policy 1 is based

solely on model uncertainty. In this policy, the probability $p_i$ that a data point $i$ is selected is proportional to the width of the 95% confidence interval for that point ($w_i$), normalized by the total width of all potential data points:

$$p_i = \frac{w_i}{\sum w}$$

Policy 1 assigns larger weights to data points with greater uncertainties. However, regions with large uncertainties are often clustered near the edges of the observed parameter space, where the GP must extrapolate far beyond observed training data [14]. However, while the uncertainty bounds of these points might be relatively high, the degree of improvement the GP can gain from sampling points from the edges of the parameter space can be limited. To avoid oversampling these areas, we developed policy 2. Policy 2 reduces the likelihood of sampling points with extreme predicted values. Specifically, the 95% confidence intervals from policy 1 are further weighted by the GP's prediction. The probability $p_i$ of selecting a data point $i$ is given by:

$$p_i = \frac{w_i \cdot ((m_{max} - m_i) \cdot m_i + 1/n)}{\sum w \cdot ((m_{max} - m) \cdot m + 1/n)}$$

where:

- $w_i$ is the 95% confidence interval width for point $i$

- $m_i$ is the GP's predicted value for point $i$

- $m_{max}$ is the maximum predicted value (3 for duration, 1 otherwise)

- $n$ is the total number of potential data points

This formulation ensures that points with high uncertainty yet with predicted values near the midpoint of the range are assigned the highest weights. We clipped the GP's predictions to the range [0, 1] for outbreak probability and $i_{max}$, and to [0, 3] for epidemic duration (since the GP predicts $\log_{10}$-transformed durations, this range corresponds to durations between 1 and 1,000 days).

For our adaptive sampling strategy, we selected 50% of the points using policy 1, and the remaining 50% using policy 2.

**Step 3: Update training data.** As mentioned earlier, the data points for $i_{max}$ and duration are based solely on simulation runs where epidemic outbreaks occurred. If a selected data point did not result in 100 outbreaks after 2,000 simulation attempts, we chose a new data point. For the initial training dataset, where no GP predictions were available, this selection was done randomly from $10^7$ LHS samples. In the active learning rounds, we chose all of the new data points as described in step 2. After successfully simulating all selected points, we added the new results to the training dataset, and a new GP training cycle began (Fig 1A).

Thanks to the optimization techniques and GPU acceleration implemented in GPyTorch [23], GP training remained computationally manageable. On our local machine (i5-12600K CPU, GeForce RTX 4090 GPU), one training round with 30,000 iterations took approximately 15 minutes to 2 hours, depending on the size of the training data (5,000 – 20,000 points, Fig 1A Step 1). Most of the computation time was spent producing new training data with the IBM during active learning rounds (Fig 1A Step 3), which could take up to ~10 hours per training round. This was especially time-intensive for the GPs modeling $i_{max}$ and outbreak duration because those required more and longer IBM runs, whereas generating data for training the outbreak probability GP was faster because many IBM simulation runs ended quickly without outbreaks.

**Gaussian process usage.** We evaluated the accuracy of the trained GP using an independent test dataset of 10,000 LHS points (Fig 1B) and calculated the RMSE. For visualizations, such as the heatmaps presented in the results section,

we clipped the predictions to the range [0, 1] for outbreak probability and $i_{max}$, and to [0, 3] for the $\log_{10}$-transformed epidemic duration.

## Sensitivity analysis

To explore how changes in parameters affect the epidemic metrics, we conducted variance-based sensitivity analyses [46] in Python using the Sobol method from the SALib library (v1.4.7) [47]. The Sobol method quantifies the contribution of single parameters and their interactions to the variance of a model's output. Since these variance components are often not analytically tractable, the Sobol method approximates them using a Monte Carlo method. The resulting sensitivity indices — first, second, and total order — provide a measure of each parameter's influence. First-order effects measure single parameter contribution, second-order effects measure the interactions of two parameters, and total order effects capture the combined impact of each parameter, including all interactions with other parameters of any order. All Sobol indices are dimensionless and normalized to the total output variance. The first-order indices sum to one only in the absence of interactions, whereas the sum of total-order indices can exceed one when interaction effects are present.

For these analyses, we used the GP predictions rather than IBM output, because estimating Sobol indices with narrow confidence intervals requires a large number of model evaluations [47], which would be computationally prohibitive using the IBM alone. Specifically, the number of model evaluations needed is proportional to $n * (2d + 2)$, where $n$ is the base sample size and $d$ is the dimensionality of the parameter space ($d = 8$; Table 1) [47]. The accuracy of the Sobol indices improves with a larger $n$, leading to smaller confidence intervals. For the sensitivity analysis of the entire input domain, where all parameters vary across their full range (Table 1), we selected $n = 2^{19}$. To investigate the first-order effect of the first case timing with the two most influential parameters (average infectivity and average mobility) held constant, we conducted a sensitivity analysis with $n = 2^{14}$ for each combination of these parameters. We calculated 95% confidence intervals of the Sobol indices using the bootstrapping method provided by SALib [47].

## Empirical data

To determine whether insights from our sensitivity analysis of the abstract GP surrogate models could inform our understanding of real-world epidemics, we analyzed over a decade of weekly dengue incidence data from Colombia [48], along with municipality-level processed demographic and environmental data from Siraj *et al.* (2018) [49]. A detailed description of the empirical data processing steps is provided in S2 Text.

We retrieved weekly dengue incidence data for Colombia from the OpenDengue database [48], covering January 1st, 2007, to December 31st, 2019, resulting in 163,279 entries. We selected this cutoff date to avoid confounding effects from the COVID-19 pandemic. To account for potential under-reporting and asymptomatic cases, we adjusted reported dengue incidences by a correction factor of 25 [50,51]. We focused on 211 municipalities with populations of at least 30,000 individuals and dengue maximum incidence rates of at least 0.1%. We defined outbreaks as periods of at least four consecutive weeks during which a smoothing spline fitted to the weekly dengue incidence exceeded the median incidence rate, resulting in the identification of 1,211 epidemic outbreaks. On average, each municipality had 6.34 outbreaks with an average outbreak duration of 195 days and an average $i_{max}$ of 0.6%.

**Parameter exploration with Gaussian processes.** We used the trained $i_{max}$ GP to explore which model parameter combinations resulted in the best agreement between predicted model output and empirical dengue incidence data from Colombia. To capture the heterogeneity in transmission potential, we incorporated municipality-specific average infectivities while keeping other model parameters (except for first case timing, but see below) constant across municipalities. Among the 173 municipalities with at least three outbreaks (1,186 epidemics total), we split the data within each municipality into 67% for calibration and 33% for testing, resulting in 737 and 449 outbreaks in each subset, respectively.

We generated 25,000 LHS from the full parameter space (Table 1), excluding the average infectivity parameter. To account for differences in incidence magnitude between simulated and empirical data, we introduced a scaling parameter (range: [0, 0.1]). This adjustment was necessary because the GP emulators were trained on IBM model outputs at daily resolution, whereas the empirical dengue data are aggregated weekly. Since the comparison with empirical data was developed after emulator training, this uniform scaling approach aligned the GP predictions with the empirical data range without requiring retraining. We applied this scaling factor uniformly across all incidence values, preserving relative differences between outbreaks and ensuring comparability across municipalities.

For each of the 25,000 LHS samples, we tested 50 evenly spaced average infectivity values in the range [0, 0.03], resulting in 1.25 million GP predictions for the 737 epidemics used for calibration. For each epidemic, the start time was adjusted via the first case timing parameter, and municipality-specific average infectivity values were used. We clipped GP predictions to the [0, 1] range before computing the RMSE between observed and predicted $i_{max}$. We then selected the average infectivity that minimized the RMSE for each of the 25,000 LHS in each municipality. We then ranked all 25,000 parameter combinations by their RMSE sums across municipalities, using the best-fit (i.e., lowest RMSE) average infectivity for each. To further investigate average infectivities across municipalities, we examined the 250 LHS combinations with the lowest RMSE sums.

Finally, we evaluated the GP's predictive performance on the withheld test data (N = 449) by calculating both the RMSE and Spearman's rank correlation coefficient ($\rho$). To test the significance of Spearman's $\rho$, we conducted 1,000 permutation tests, where the start time and municipality for each epidemic in the test set were randomly shuffled.

## Statistical analysis

Unless stated otherwise, we performed statistical analyses using the R statistical computing environment (v4.2.1) [52]. We declared significance at an alpha cut-off of 5%.

## Results

### Gaussian process performance

To enable a more efficient exploration of the output space of our IBM, we trained GP surrogate models on input-output data pairs from the IBM. Specifically, we trained three independent GPs to predict outbreak probability, $i_{max}$, and outbreak duration.

**Runtime.** Once trained, the GPs predictions were almost instantaneous: with our local machine, we were able to generate about 100,000 predictions per second. Because each GP prediction represents a metric calculated and averaged over 100 IBM simulation runs, 100,000 GP predictions are equivalent to performing at least $10^7$ IBM simulations, which would typically require several hundred CPU hours. The efficiency of the GPs originates from the closed-form nature of their predictions: computationally expensive matrix inversions are performed only during training [14,22]. As a result, prediction runtimes are deterministic and – in contrast to the underlying IBM – independent of outbreak duration.

**Accuracy.** We evaluated model performance using RMSE — both for checks against a validation dataset during training to avoid overfitting (Fig 1A), and for assessing the accuracy of the GPs after training was completed (Fig 1B). During training, we observed that the first few adaptive training rounds tended to lead to the most significant improvements in model performance, whereas additional rounds later in the process yielded diminishing returns (S2 Fig). The final GPs achieved RMSE values of 0.058 for outbreak probability, 0.042 for $i_{max}$, and 0.068 for duration (S2 Fig).

We observed greater variance in the model's predictive accuracy for weaker epidemics (i.e., lower $i_{max}$ values) and longer epidemics (i.e., larger duration) (Fig 2). In such cases, stochasticity played a larger role, making predictions more challenging. For the $i_{max}$ GP, there were instances where the intensity of the epidemic outbreak was severely overpredicted (Fig 2B). This might have resulted from neighboring data points with very different properties. Although our kernel choice

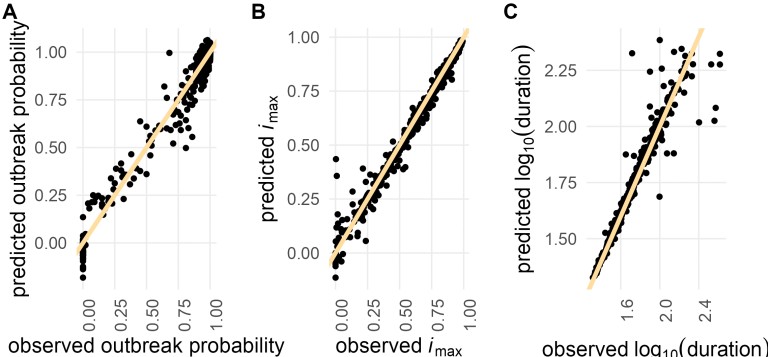

**Fig 2. Gaussian Process performance evaluation.** Comparison of observed versus predicted values for 500 randomly sampled test data points. The yellow line represents the identity line ($x=y$) for (A) outbreak probability (B) maximum incidence ($i_{max}$), and (C) $\log_{10}$-transformed duration.

allows for rather abrupt changes in function values, the interpolation might not fully capture the true dynamics of the underlying model if it does not happen exactly at the midpoint between the two points.

## Sensitivity analysis & specific model outcomes

The GP surrogate models enabled comprehensive exploration of the parameter space at a fraction of the computational cost of the IBM. This allowed us to perform detailed global sensitivity analyses to identify the parameters and interactions most strongly influencing epidemic outcomes. We used the trained GPs to conduct variance-based Sobol sensitivity analyses [9] to quantify the contributions of individual parameters and their interactions to the overall variance of the model output. We estimated first-order effects due to individual parameters, second-order effects due to pairwise interactions between parameters, and total-order effects that include all first- and second-order interactions as well as all higher-order interactions.

**Entire input domain.** We observed that average infectivity and average mobility are the primary drivers in our model, shaping all three epidemiological metrics: outbreak probability, $i_{max}$, and outbreak duration. Since there is no correlation between the number of visits sampled for a given individual over time, our model does not include systematic superspreading behavior. As a result, we expected the sensitivity index for mobility skewness to be low across all three metrics, in contrast to a stronger impact of average mobility. Our findings confirmed this expectation, with mobility skewness showing no influence on the epidemiological metrics ($i_{max}$: Fig 3A, outbreak probability: S3A Fig, outbreak duration: S4A Fig).

The first-order effect estimates for average infectivity are nearly identical across all three metrics: outbreak probability, $i_{max}$, and duration (0.52, 0.53, and 0.53, respectively; Figs 3, S3, and S4). However, the total effect of average infectivity is notably higher for outbreak probability (0.69; S3 Fig) compared to $i_{max}$ and duration (both 0.58; Figs 3 and S4). The higher total-order effect for outbreak probability is primarily driven by the interaction between average infectivity and average mobility (S3B – S3C Fig). This makes intuitive sense, as highly infectious diseases can still trigger outbreaks even when individual movement is limited. By contrast, the spread of less infectious diseases relies more heavily on sufficient individual mobility to compensate for a lower per-contact infection probability. Accordingly, the first-order effects of average mobility are smaller for outbreak probability compared to $i_{max}$ and duration (0.12 vs. 0.26 and 0.27). However, the total-order effects of average mobility are similar across all three metrics (0.27, 0.29, and 0.3, respectively, Figs 3, S3, and S4).

The reduced importance of the interaction between average infectivity and average mobility for $i_{max}$ and outbreak duration is due to the fact that these metrics are calculated only across simulations in which an actual outbreak occurred. For these two metrics, the largest second-order effect is the interaction between the seasonality strength and the timing of the

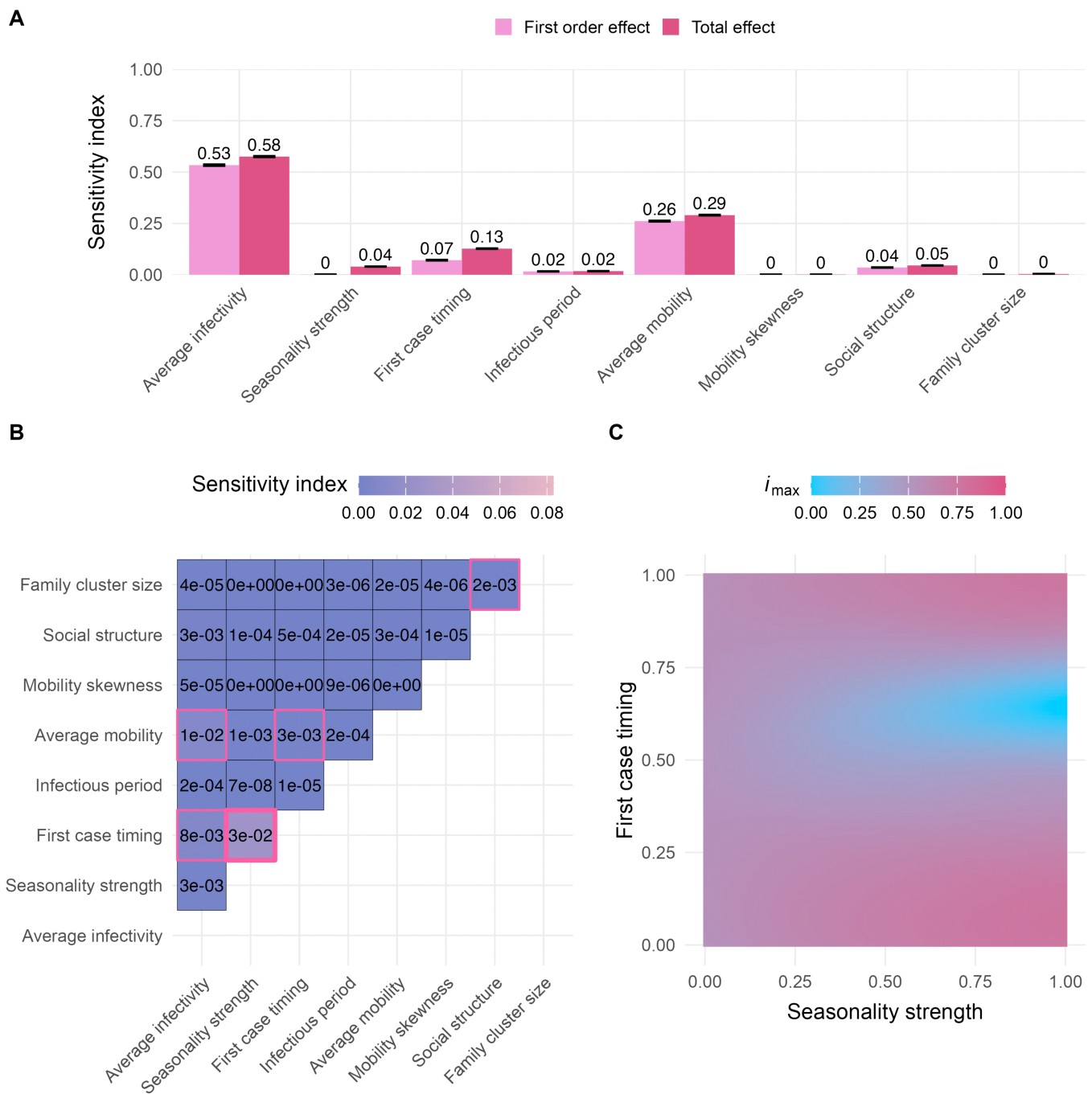

**Fig 3. Sobol sensitivity analysis, maximum incidence ($i_{max}$).** (A) First-order and total effects across the entire input domain (Table 1). The first-order effect describes the impact of a single parameter on the model output ($i_{max}$), while the total effect of a parameter accounts for both its first-order effect and all interactions with other parameters. Error bars represent the 95% confidence intervals of the sensitivity index estimates. We evaluated a total of 9,437,184 points for the sensitivity analysis. (B) Second-order effects across the entire input domain (Table 1). A second-order effect captures the pairwise interaction between two parameters. Sobol indices with a 95% confidence interval that does not overlap zero are highlighted with a pink border. The largest second-order effect is emphasized with a bold pink border. (C) $i_{max}$ predictions with varying seasonality strength and first case timing parameters (i.e., the two parameters with the largest second-order effect, see panel B). Other parameters were fixed at default values (Table 1). Corresponding Sobol sensitivity analysis plots for outbreak probability and outbreak duration can be found in S3 and S4 Figs.

first infectious case (Figs 3B and S4B). In our model, the infection probability fluctuates seasonally, following a cosine pattern, and thus the initial infection timing relative to the seasonal cycle is crucial. Introducing the disease during a low-risk season (i.e., when the value of the first case timing parameter falls between 0.25 and 0.75) can lead to prolonged epidemics with lower $i_{max}$ (Figs 3C and S4C). Since the infection probability depends on the interaction between the average infectivity, the seasonality strength, and the first case timing, it is encouraging that sensitivity analyses of the GP surrogate models effectively uncovered these pairwise interactions (Figs 3B, S3B, and S4B).

We also observed that family cluster size has only a minor effect on the outcomes, which is mainly driven by interactions with the social structure parameter (Figs 3B, S3B, and S4B). Since individuals return home each day, they are more likely to interact with others within their home location and family cluster (as long as the social structure parameter is > 0). Family cluster size determines how many individuals, on average, live within each family cluster, and in combination with social structure, it influences the extent of interaction within those family clusters, thus having some effect on the investigated model outputs.

**Conditional parameter subdomains.** The results of variance-based sensitivity analyses depend on the variance present in the model output: when a few parameters account for a disproportionately large amount of variance in model output, the contributions of other parameters can be difficult to detect. For our results, this was the case for average infectivity and average mobility (Figs 3, S3, and S4). To address this, we conducted additional sensitivity analyses in subdomains of the parameter space, where average infectivity and average mobility were each fixed at selected values. This allowed us to estimate the Sobol indices for seasonality strength and first case timing on outbreak probability within these parameter subdomains.

Interestingly, we observed a sharp increase in the first-order indices for low average infectivity values, followed by a gradual decline as the average infectivity increased (Fig 4A). This pattern corresponds to a shift in the system: moving from a state where epidemic outbreaks are rare (Fig 4B first panel) to one where outbreaks occur in the majority of simulated scenarios (Fig 4B fourth panel). Under conditions that are generally unfavorable for disease transmission, outbreaks are rare and occur only when all parameters align to support an outbreak. Consequently, the first-order sensitivity indices for the first case timing parameter tend to be low, because this metric reflects only the independent effects of single parameters. With increasing average infectivity, the system enters a state where the outbreak probability is primarily driven by the first case timing, creating a hit-or-miss dynamic (Fig 4B third panel). Here, the strength of seasonality plays a minimal role; the key factor determining whether an outbreak occurs is whether the first case is introduced when infection probabilities are above or below the average infectivity. At higher average infectivity levels, the outbreak probability heatmap displays a U-shaped pattern, with low predicted outbreak probabilities when first case timing coincided with periods unfavorable for transmission (Fig 4B fourth panel), indicating that seasonality strength again became a critical parameter. With further increases in average infectivity, the proportion of unlikely outbreak scenarios shrinks, and the first-order sensitivity index of first case timing gradually declines. We confirmed the GP-predicted outbreak probability patterns by conducting an additional set of simulations with the IBM, verifying that these results are not artifacts of the surrogate model. (Fig 4C).

## Application to empirical dengue incidence data

To assess whether insights from our sensitivity analysis translate to real-world epidemics, we analyzed over a decade of municipality-level dengue incidence data from Colombia [48,49]. We first tested our model's prediction of an inverse relationship between average infectivity and average human mobility (S3C Fig). Due to the lack of fine-scale data, we used mosquito abundance probability [49,53] as a proxy for average infectivity. While practical, this simplification does not capture the full complexity of real-world transmission dynamics, which are also influenced by factors such as vector control, urbanization, and human-mosquito contact patterns. For average human mobility, we used the inverse of mean travel time as a proxy [49,54], where mean travel time represents the average duration required to reach a settlement with at

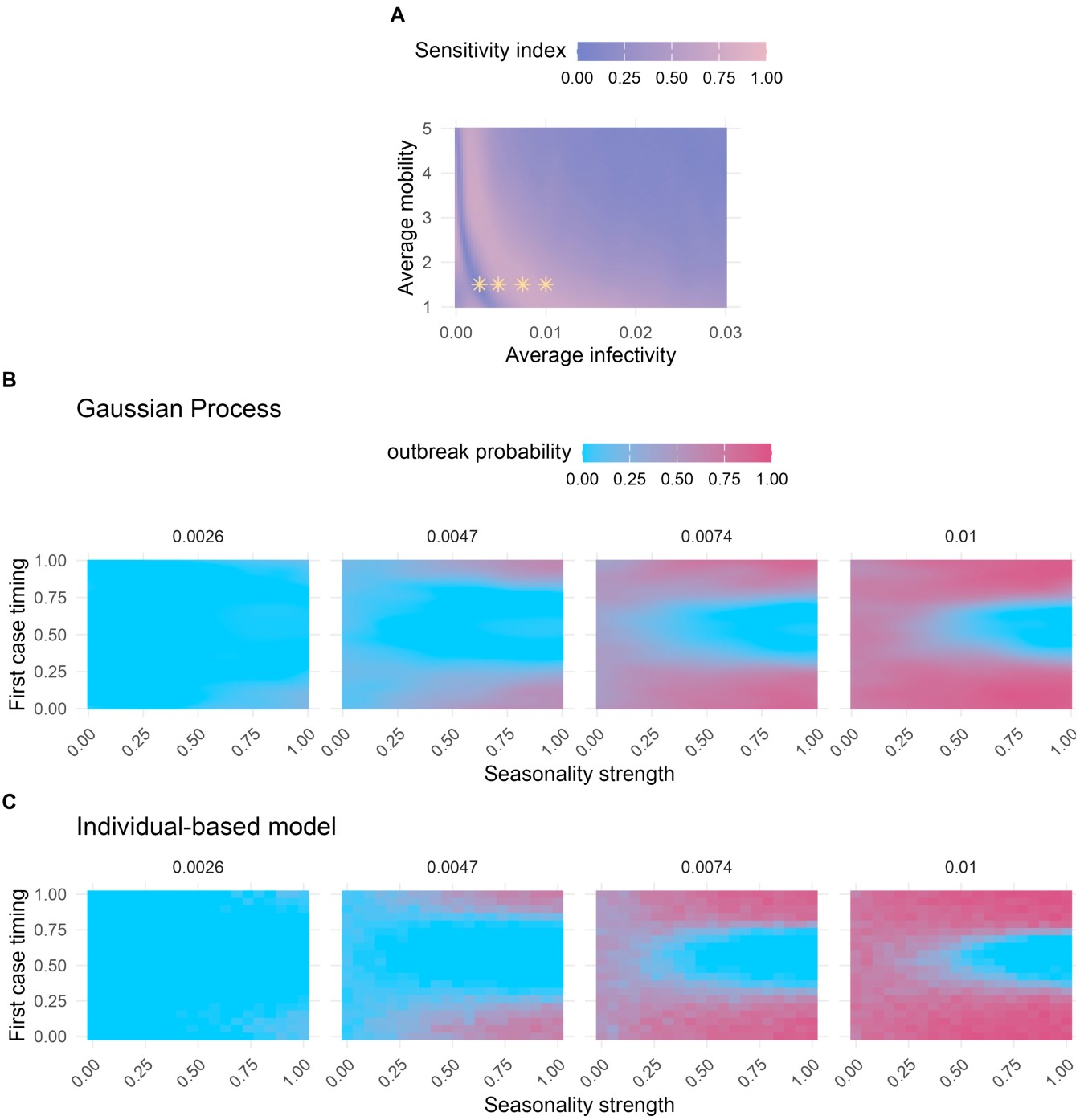

**Fig 4. Summary of model outcomes related to outbreak probability.** (A) First-order sensitivity index estimates for the first case timing parameter across varying average infectivity and average mobility values. For each parameter combination, we evaluated a total of 294,912 points. We varied all other parameters across their full ranges (Table 1). The first-order effect measures the influence of a single parameter on the model output (outbreak probability). Yellow stars mark parameter combinations associated with specific model outcomes shown in (B). (B) Predicted outbreak probabilities using the Gaussian Process surrogate model with varying seasonality strength and first case timing values. Panels represent different average infectivities. All other parameters were fixed at default values (Table 1), except for average mobility, which was set to 1.5. (C) Outbreak probabilities inferred from the individual-based model, with varying seasonality strength and first case timing values. Panels represent different average infectivities. As in (B), the remaining parameters were fixed at default values (Table 1), except for average mobility which was set to 1.5. (B) and (C) thus represent model outcomes for the same model parameters, but conducted with the Gaussian Process surrogate model (B) versus the original individual-based model (C), allowing a direct comparison between the two.

least 50,000 inhabitants (not necessarily within the same municipality). Thus, mean travel time primarily reflects regional connectivity rather than local within-municipality movement, and may not fully capture all nuances of mobility relevant to disease transmission.

From our sensitivity analysis, we expected a positive correlation between mean travel time and mosquito abundance for real-world epidemic outbreaks. However, we found no significant positive correlation, even when restricting the analysis to most remote municipalities (travel time $>=$ 85th percentile; Spearman's rank correlation test: $\rho = 0.1$, $S = 917,546$, $p = 0.085$). This lack of correlation likely reflects the limitations of mean travel time as a proxy for the kind of human mobility that drives local epidemic dynamics.

We next tested our model's prediction that the timing of the first infectious case strongly influences outbreak dynamics when seasonality is important. Empirical data, however, only captures measurable outbreaks in the population; we lack information on the actual introduction of the first case or on instances where an introduction of an infectious individual did not result in an epidemic outbreak. Binning the observed outbreaks by week, we found that their distribution is not uniform throughout the year (Chi-squared test: $\chi^2 = 117.85$, df $= 52$, $p < 0.001$), indicating a seasonal effect consistent with our expectations for dengue outbreak dynamics [50,55].

To evaluate how well our calibrated $i_{max}$ GP can capture real-world dengue outbreaks, we used it to identify parameter combinations that best reproduced observed epidemic outbreaks in Colombian municipalities. The parameter combination that minimized the RMSE on the calibration data revealed a high degree of social structure in the model (seasonality strength $= 0.16$; first case timing $= 0.58$; infectious period $= 4.67$; average mobility $= 4.39$; mobility skewness $= 0.47$; social structure $= 0.99$; family cluster size $= 4.54$; scaling factor $= 0.03$). Across the top 1% (250 out of 25,000) parameter combinations with the lowest RMSE, however, we observed a wide range of values (S1 Table), often spanning the entire parameter range, indicating that multiple distinct parameter combinations can produce similar epidemic patterns in our IBM.

Calibrated average infectivity estimates also varied across municipalities, though certain municipalities consistently exhibited higher average infectivity estimates across the top-ranked parameter combinations (i.e., those with the lowest RMSE; Fig 5A). Most of these municipalities overlapped with previously reported dengue disease clusters [57] and had a significantly higher Gross Cell Product compared to other municipalities (Fig 5B; *Wilcoxon rank sum test: W = 1,037, p = 0.021*), suggesting a potential link between economic activity [56] and dengue incidence.

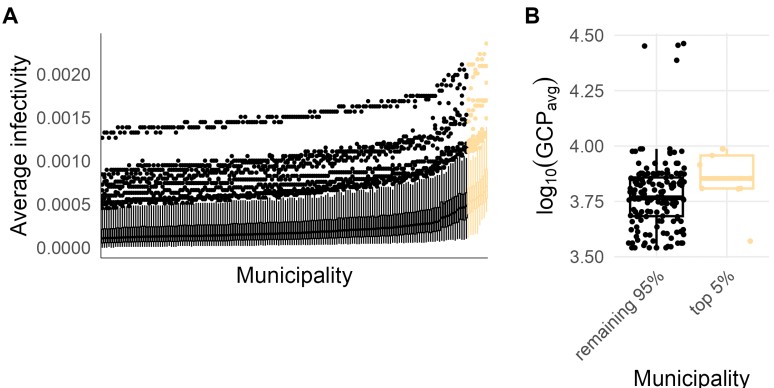

**Fig 5. Municipality-level average infectivity estimates and Gross Cell Products.** (A) Distribution of municipality-specific average infectivity estimates for the 250 parameter combinations with the lowest root mean square errors. The top 5% of municipalities as sorted by median average infectivity estimates are highlighted in yellow. (B) Average $\log_{10}$-transformed Gross Cell Product (GCP) — a measure of economic activity [56] where higher values represent greater economic activity — distributions, as reported by Siraj *et al.* (2018), for the municipalities depicted in (A), with the municipalities with the largest average infectivity estimates grouped separately.

Finally, we evaluated the predictive performance of the calibrated $i_{max}$ GP using the best-fit parameters and withheld test data (N = 449). While the model achieved an RMSE of 0.006, the normalized RMSE (the RMSE scaled by the mean of the observed data) was 1.02, indicating that the model struggled to capture the full complexity of the system and that the model's predictions were not highly accurate (S5A Fig). The rank correlation coefficient between observed and predicted values was 0.458, and permutation tests placed the model in the top percentile, demonstrating modest predictive power (S5B Fig). However, these results underscore the limitations of our approach to large-scale, heterogeneous epidemic data.

## Discussion

In this paper, we demonstrated the potential of statistical emulation for studying the dynamics of epidemiological IBMs. Specifically, we implemented an abstract individual-based disease transmission model, loosely inspired by dengue, in C++ and trained Gaussian Process (GP) emulators to approximate three key outbreak metrics: outbreak probability, maximum incidence ($i_{max}$), and outbreak duration. Due to their fast prediction speed, these GPs facilitated highly efficient exploration of the model's eight-dimensional parameter space, allowing us to conduct comprehensive sensitivity analyses that would have been computationally prohibitive using the IBM directly. Our results show that average infectivity and average mobility have large first-order effects and influence all three epidemiological metrics. The most important pairwise parameter interaction varies by model outcome: the interaction between average infectivity and the average human mobility primarily influences outbreak probability, whereas the timing of the first infectious case, combined with seasonality strength, can shape both $i_{max}$ and the duration of epidemics. Although our trained GP — and the underlying IBM — do not fully capture the full complexity and heterogeneity of real-world dengue dynamics, they provide a computational efficient framework for exploring broad epidemiological patterns and trends. When applied to Colombian dengue incidence data, the approach highlighted municipalities that overlap with previously identified dengue clusters [57], illustrating how statistical emulation can complement empirical research by linking computational modeling with observed disease distributions.

### Individual-based model

The primary aim of this study was to demonstrate the potential of GP emulation as a tool for efficiently analyzing individual-based epidemiological models, rather than to construct a detailed, disease-specific representation of dengue. However, because we applied the framework to Colombian dengue data for illustration, it is important to acknowledge key simplifying assumptions in our IBM relative to known dengue disease transmission characteristics. For example, we modeled an initially completely susceptible population in our simulations, neglecting any preexisting immunities at the onset of the epidemic. Our approach ignores the fact that dengue is caused by four distinct viral serotypes (DENV–1 to DENV–4), and while infection with one strain provides long-lasting immunity against that specific strain, immunity to other strains lasts only a short time [58]. Moreover, a second infection with a different serotype can trigger antibody-dependent enhancement, significantly increasing the risk of severe (and symptomatic) dengue [58]. In hyperendemic countries such as Colombia [50], where multiple dengue virus serotypes are simultaneously circulating within the population, this can cause complex immunity dynamics. Unfortunately, strain-specific sequencing data and antibody measurements that could be used to accurately estimate the proportion of immune individuals are scarce [50].

Furthermore, while our abstract IBM incorporates some key aspects of dengue epidemiology, such as the infectious duration in humans [37] and the role of human movement [38,44], we chose not to explicitly model mosquito vectors. Combining host models with detailed vector models that account for factors such as habitat availability and selection pressures across mosquito life stages could significantly enhance the realism of epidemiological simulations [2,59], albeit at a substantial cost in model complexity, number of parameters, and simulation runtime.

Another simplifying assumption in our IBM is in the human mobility model. While the family cluster size and social structure parameters allow us to model populations with varying levels of social interconnectivity, locations are not

spatially explicit, meaning that the distance between them is not defined. Thus, the likelihood of a person visiting a location is solely determined by parameters affecting social population structure and human mobility. Real-world human movement patterns, on the other hand, are known to exhibit strong spatial regularity [8,60,61]. Moreover, in reality, human populations are rarely closed systems like the one we modeled here. Migration and a variety of factors — economic shifts, environmental changes, large public events — often lead to interactions beyond regular social circles, increasing the risk of disease introduction into areas that were previously unaffected [62].

### Gaussian processes

While we decided to train our GPs on outbreak probability, $i_{max}$, and duration, a GP could instead be trained on other outputs from the IBM. For example, a GP could be trained on the total epidemic size or the time to the epidemic peak, if relevant to addressing the research question at hand. It would also be possible to use a so-called multi-task GP [63], which allows the simultaneous prediction of multiple outputs, and is capable of capturing correlations between them. This could improve the efficiency of the training process, especially when the outputs are highly correlated, because multi-task GPs can leverage shared information between the prediction tasks to enhance accuracy and reduce computational costs. Our choice of separate GPs was guided by two factors. First, we trained the outbreak probability GP on the proportion of simulation runs with observed outbreaks, whereas we trained the $i_{max}$ and duration GPs exclusively on simulations with observed outbreaks, which made choosing a consistent set of training points across all three metrics challenging. Second, we had no clear expectations regarding the correlation between $i_{max}$ and outbreak duration: simulations with shorter durations might result from severe epidemics where most individuals are infected rapidly (high $i_{max}$), or from scenarios where the disease quickly dies out (low $i_{max}$). These complex dynamics made separate GPs a simpler, more practical choice.

A key factor in implementing a GP is the choice of an appropriate kernel [14,22]. We used the Matérn kernel because of its flexibility in modeling different levels of smoothness in the data. For this kernel, we chose a smoothness parameter $v = 0.5$, which can be beneficial for capturing model behavior in which small changes in parameters can result in abrupt changes in model outputs, as seen here. Preliminary testing, as well as our trained GPs, showed satisfactory performance with the Matérn kernel, so we did not pursue alternative kernels. Whether the accuracy of our GPs could be improved even further with more customized or composite kernels tailored to specific features of the data remains to be explored.

One key advantage of GPs is their Bayesian nature, which allows for uncertainty quantification. This property is particularly useful in active learning, wherein the uncertainty measurements can be leveraged to choose the most informative points to add to the training data. During GP training, we selected half of the new points based on the confidence interval widths, while the other half was selected using the product of the confidence interval widths and a function of the predicted mean. Specifically, we weighted the confidence interval widths based on how close the predicted mean was to its most extreme possible values, assigning the highest weights to intermediate predictions. This approach encourages the GPs to move away from the edges of the parameter space, where uncertainties are naturally higher and predicted means often become extreme. These extremes occur either due to expected model behavior at the parameter boundaries (extreme parameter values cause extreme model behavior), or because data is sparse in these regions, causing the GP to revert to its prior (a constant mean of 0 in our case, which is an extreme value relative to the average predicted value) [22]. However, this approach might overlook regions that the GP does not determine to be highly uncertain, but which could provide valuable information if explored. Alternative sampling strategies, such as expected improvement, could help identify points that boost model performance, even if their initial uncertainty is lower. Moreover, tools like BoTorch [64] provide libraries to implement advanced batch optimization techniques, allowing the selection of sets of data points that are chosen together to maximize their combined impact on improving GP performance. While more advanced techniques like expected improvement scores and batch optimization could potentially enhance GP performance, they would require further model tuning and validation, which is beyond the scope of this study.

   

Finally, we would like to point out that GPs are only one of numerous possible choices for a surrogate model. Alternative machine learning approaches — such as random forests, support vector machines, and neural networks — can also be used to build effective surrogate models for complex IBMs. Neural networks, in particular, are well suited for capturing highly nonlinear or erratic model behavior, especially when modern computational resources such as GPUs are available [65]. We chose GPs in this study because they are a well-established emulation technique that provides a solid probabilistic foundation allowing uncertainty quantification [14]. This uncertainty quantification, in turn, enables efficient active learning strategies for selecting additional training data. Implementing such active learning strategies is considerably more straightforward with GPs than with neural networks, for which obtaining reliable uncertainty estimates is more challenging. Nevertheless, previous work has shown that neural networks can outperform GPs in predictive accuracy for highly nonlinear systems [13]. For applied emulation tasks, researchers must therefore choose the method that best fits their simulation model's characteristics and the available computational resources.

### Sensitivity analysis

The fast prediction speed of the trained GPs allowed us to conduct comprehensive variance-based sensitivity analyses. However, this approach could be confounded by potential discrepancies between the GP surrogate model and the original IBM. While the GPs generally predicted epidemiological metrics inferred from the IBM well, the width of the sensitivity analysis confidence intervals should be interpreted cautiously. Furthermore, average infectivity and average mobility emerged as the dominant contributors to variance in the epidemiological metrics, making it harder to detect the influence of the other parameters. This scaling effect can obscure smaller, but still relevant, factors. To address this, we also performed sensitivity analyses in targeted regions of the parameter space for which we fixed average infectivity and average mobility, revealing state changes within the model's dynamics — sudden transitions from rare epidemic outbreaks to frequent outbreaks — which we confirmed by simulating selected points directly with the IBM. However, it is important to note that while a sensitivity analysis captures the variance in model outputs due to parameter changes, it does not fully capture the underlying dynamics of the model, such as state transitions or the mechanistic interactions between single parameters that drive these changes. Specifically, the sensitivity analysis highlights which parameters contribute most to the output variance, but it does not reveal why certain parameter combinations lead to changes in model behavior.

We observed the largest second-order effects between average infectivity and average mobility for the outbreak probability metric, and between seasonality strength and first case timing for $i_{max}$ and duration. To explore how well these model-derived insights translate to real-world epidemic outbreaks, we examined over a decade of dengue incidence data from Colombia. We used mosquito abundance probability [49,53] as proxy for average infectivity, implicitly assuming a constant biting rate whereby higher mosquito abundance directly translates to increased infection probability. However, this simplified representation overlooks the complexity of real-world disease transmission dynamics, which is shaped by factors such as vector control [66], urbanization, human-mosquito contact rates [67], and mosquito behavior [68]. We did not observe the expected correlation between our proxies for average human mobility and average infectivity in the empirical outbreak data. This may be partly due to previous findings that mean travel time serves as a broad indicator of accessibility rather than a precise measure of actual human mobility [49]. While our analysis supports a seasonal pattern of dengue outbreaks consistent with prior studies [50,55], we treated each epidemic as independent, not accounting for temporal correlations within municipalities or spatial dependencies across neighboring or well-connected municipalities. As a result, the statistical significance of our findings should be interpreted cautiously. Overall, these results illustrate both the potential and the current limitations of applying abstract individual-based model insights — based on a simplified disease transmission framework — to empirical epidemiological data.

## Application to empirical dengue incidence data

When comparing the predictions of our model with real-world outbreaks, the only empirical parameter that actually varied between epidemics within each municipality was the epidemic's onset. This limited the GP's flexibility to generate diverse predictions within each municipality. In fact, a simple linear mixed-effects model that predicts $\log_{10}$-transformed $i_{max}$ values based on the onset timing of an epidemic, while accounting for the municipality-level variations with random effects, performed similarly to the GP model on withheld test data (Spearman's $\rho = 0.53$). This suggests that both the abstract IBM and the GP emulators might be too generalized to effectively predict real-world outbreak data across multiple municipalities. To achieve more accurate predictions, the IBM would need to be more complex, incorporating municipality- and disease-specific characteristics such as outbreak histories, population immunity levels, and finer-scale human movement patterns, which might be critical for capturing the nuanced dynamics of local outbreaks.

Despite the GP emulator's (and the underlying IBM's) limitations in capturing the full complexity of empirical dengue dynamics, our analysis revealed that a subset of municipalities consistently exhibited higher average infectivity estimates. Several of these municipalities — such as Puerto López, Leticia, Melgar, and La Mesa — stand out due to their economic or geographic context. For example, Puerto López, which had the highest average infectivity estimate, is a key river port, while Leticia is located at the tri-border area of Colombia, Brazil, and Peru, functioning as a major hub on the Amazon river. Tourist destinations such as Melgar and La Mesa also showed elevated average infectivity estimates, potentially reflecting increased human movement and connectivity driven by tourism and travel — factors that may enhance dengue transmission in these areas. These observations support the idea that human movement and economic activity could play a significant role in shaping dengue dynamics [69].

At the same time, municipalities with higher average infectivity estimates also tended to have greater economic activity, which may be associated with better healthcare access and, in turn, increased detection and reporting of dengue cases. This introduces potential bias, because our model assumes constant reporting rates across municipalities, highlighting the need for caution when interpreting these findings. Nonetheless, many of the municipalities with elevated average infectivity estimates are located in areas previously identified as disease clusters for dengue and other *Aedes*-borne diseases [57].

## Conclusion

In conclusion, we explored the utility of statistical emulation to efficiently analyze epidemiological IBMs. The use of GP-based emulators allowed valuable insights into the key drivers of our simulated disease dynamics, revealing critical interactions between average infectivity, human mobility, and seasonality. Overall, our work demonstrates both the potential and the challenges of using statistical emulation to explore complex epidemiological systems, providing a foundation for future efforts that could incorporate additional model complexity and realism while maintaining computational efficiency.

## Supporting information

**S1 Text. Detailed individual-based model description.**
(PDF)

**S2 Text. Detailed empirical data processing description.**
(PDF)

**S1 Fig. Schematic overview of human movement in the individual-based model.** Each colored frame represents a unique, non-overlapping family cluster, with each cluster containing multiple family homes. Individuals can make visits within their own family cluster (solid arrows) or to other clusters (dashed arrows). The likelihood of visits occurring inside the family cluster is determined by the social structure parameter (Table 1). Each individual visits their home at least once

per day and moves independently of others in the same family (individuals A and B). Multiple visits to the same location are allowed (individual D). Visits to other family clusters occur randomly and are not restricted to any specific cluster (individual C).
(PNG)

**S2 Fig. Validation RMSE between Gaussian Process predictions and individual-based model results (N = 10,000).** The Root mean squared error (RMSE) decreases as the size of the dataset used to train the Gaussian Processes increases (x-axis). The RMSE between the predictions of the final GP model and the test data (N = 10,000 data points) is indicated by a yellow square. (A) outbreak probability (B) maximum incidence ($i_{max}$), (C) $\log_{10}$-transformed duration.
(PNG)

**S3 Fig. Sobol sensitivity analysis, outbreak probability.** (A) First-order and total effects across the entire input domain (Table 1). The first-order effect describes the impact of a single parameter on the model output (outbreak probability), while the total effect accounts for all interactions involving one or more parameters. Error bars represent the 95% confidence intervals of the sensitivity index estimates. We evaluated a total of 9,437,184 points for the sensitivity analysis. (B) Second-order effects across the entire input domain (Table 1). A second-order effect captures the pairwise interaction between two parameters. Sobol indices with a 95% confidence interval that does not overlap zero are highlighted with a pink border. The largest second-order effect is emphasized with a bold pink border. (C) Predicted outbreak probabilities with varying average infectivity and average mobility parameters (i.e., the two parameters with the largest second-order effect, see panel B). Other parameters were fixed at default values (Table 1).
(PNG)

**S4 Fig. Sobol sensitivity analysis, $\log_{10}$-transformed duration.** (A) First-order and total effects across the entire input domain (Table 1). The first-order effect describes the impact of a single parameter on the model output ($\log_{10}$(duration)), while the total effect accounts for all interactions involving one or more parameters. Error bars represent the 95% confidence intervals of the sensitivity index estimates. We evaluated a total of 9,437,184 points for the sensitivity analysis. (B) Second-order effects across the entire input domain (Table 1). A second-order effect captures the pairwise interaction between two parameters. Sobol indices with a 95% confidence interval that does not overlap zero are highlighted with a pink border. The largest second-order effect is emphasized with a bold pink border. (C) $\log_{10}$(duration) predictions with varying seasonality strength and first case timing parameters (i.e., the two parameters with the largest second-order effect, see panel B). Other parameters were fixed at default values (Table 1).
(PNG)

**S5 Fig. Comparison of observed and predicted maximum incidence and correlation analysis across randomized permutations.** (A) Observed vs. predicted maximum incidence ($i_{max}$) for empirical epidemic outbreaks (N = 449). The yellow line represents the identity line ($x = y$). (B) Distribution of Spearman correlation coefficients between observed and predicted $i_{max}$ from 1,000 permutations, where both the onset and municipality of the 449 epidemics were randomized. The actual observed correlation coefficient is shown as a vertical yellow line.
(PNG)

**S1 Table. Summary of 250 parameter sets with lowest RMSE from parameter exploration with Gaussian Process.** Summary statistics describing the 250 parameter combinations with the lowest root mean squared errors from the parameter exploration with the Gaussian Process. These combinations represent the best-fitting sets of parameters for matching observed and predicted dengue maximum incidences across municipalities. IQR = Interquartile range (range between the 25th and 75th percentiles).
(PDF)

## Acknowledgments

We thank all members of the Messer and Murdock lab for helpful discussions. Special thanks to Beliz Erdogmus for her contributions during the early phases of the project; Isabel Kim, Mitchell Lokey, and Meera Chotai for technical support; Amir Siraj for support with the municipality-specific environmental data; and Oliver Brady for providing supplemental shape files.

## Author contributions

**Conceptualization:** Anna Maria Langmüller, Courtney C. Murdock, Philipp W Messer.

**Data curation:** Anna Maria Langmüller, Benjamin C. Haller.

**Formal analysis:** Anna Maria Langmüller, Kiran A. Chandrasekher.

**Funding acquisition:** Anna Maria Langmüller, Philipp W. Messer.

**Investigation:** Anna Maria Langmüller, Courtney C. Murdock, Philipp W. Messer.

**Methodology:** Anna Maria Langmüller, Benjamin C. Haller, Samuel E. Champer, Philipp W. Messer.

**Software:** Anna Maria Langmüller, Benjamin C. Haller, Samuel E. Champer.

**Validation:** Anna Maria Langmüller, Benjamin C. Haller.

**Visualization:** Anna Maria Langmüller, Kiran A. Chandrasekher.

**Writing – original draft:** Anna Maria Langmüller, Philipp W. Messer.

**Writing – review & editing:** Anna Maria Langmüller, Kiran A. Chandrasekher, Benjamin C. Haller, Samuel E. Champer, Courtney C. Murdock, Philipp W. Messer.

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
