## [Decision Letter · Decision Letter 0]

30 Sep 2025

Gaussian Process emulation for exploring complex infectious disease models

PLOS Computational Biology

Dear Dr. Langmüller,

Thank you for submitting your manuscript to PLOS Computational Biology. After careful consideration, we feel that it has merit but does not fully meet PLOS Computational Biology's publication criteria as it currently stands. Therefore, we invite you to submit a revised version of the manuscript that addresses the points raised during the review process.

Please submit your revised manuscript within 60 days Nov 30 2025 11:59PM. If you will need more time than this to complete your revisions, please reply to this message or contact the journal office at ploscompbiol@plos.org. Please include the following items when submitting your revised manuscript:

We look forward to receiving your revised manuscript.

Kind regards,

Jennifer A. Flegg

Section Editor

PLOS Computational Biology

Jennifer Flegg

Section Editor

PLOS Computational Biology

**Journal Requirements:**

At this stage, the following Authors/Authors require contributions: Anna M Langmüller, Kiran A Chandrasekher, Benjamin C Haller, Samuel E Champer, Courtney C Murdock, and Philipp W Messer. Please ensure that the full contributions of each author are acknowledged in the "Add/Edit/Remove Authors" section of our submission form.

4) We notice that your supplementary Figures, and Tables are included in the manuscript file. Please remove them and upload them with the file type 'Supporting Information'. Please ensure that each Supporting Information file has a legend listed in the manuscript after the references list.

5) Please ensure that the funders and grant numbers match between the Financial Disclosure field and the Funding Information tab in your submission form. Note that the funders must be provided in the same order in both places as well.

State what role the funders took in the study. If the funders had no role in your study, please state: "The funders had no role in study design, data collection and analysis, decision to publish, or preparation of the manuscript.".

**Reviewers' comments:**

Reviewer's Responses to Questions

**Comments to the Authors:**

Reviewer #1: The authors present a well-written paper demonstrating the use of Gaussian Process emulators for exploring the parameter spaces of epidemiological models. As a frequent user of GPs for my own ABM-centric work, I'm pleased to see another demonstration of the power and utility of this approach for the exploration of complex models; however, it's worth noting that other machine-learning methods can provide better fits for complex ABMs than GPs (see, e.g., Angione, Claudio, Silverman, Eric and Yaneske, Elisabeth (2022). Using machine learning as a surrogate model for agent-based simulations. PLoS ONE, 17(2), e0263150. (doi: 10.1371/journal.pone.0263150).

I find the authors' use of individual-based models (IBMs) as an overarching category to be somewhat confusing. In my career as a complex systems scientist focussed on ABMs, a distinction is normally drawn between IBMs that use transition probabilities to determine changes in agent state (e.g., microsimulations), and those that use decision rules (ABMs), with the latter being considered significantly harder to analyse. I believe the introduction to the paper would benefit from expressing more clearly the differences between individual-based and agent-based approaches, particularly for those audiences in complex systems disciplines where IBM is not a commonly used term.

Other than that, the benefits of using GPs and exploring a model's parameter space more generally are explained well and with sufficient detail. I would only add here that other methods of emulating complex models can provide better emulator fit, assuming appropriate computing hardware is available; for example, neural networks are excellent at replicating models with highly non-linear behaviour, but benefit from GPUs in order to provide sufficient time savings to make the method worth the effort.

The IBM presented here I would class as a microsimulation-style simulation, as the movements of the agents and the disease dynamics are apparently generated using parameters drawn from distributions rather than decision rules driving the individuals. In other words, there is top-down imposition of parameters here, which precludes the model from being an ABM, where decision rules and agent heterogeneity are used to drive population-level patterns from the bottom up. That being the case, I hope the authors can make more clear where their model sits on the IBM-ABM spectrum.

The results are well-written and well-presented. I'm somewhat surprised that the GPs took quite so long to run for such a simple model; my own experiences with this approach has pegged GPs as being among the fastest available emulation techniques, typically finishing ~40-50K emulated runs in a few minutes, without the use of GPUs. That being said, I'm pleased that the authors have made the effort to develop and share their own tools, which is always of benefit to the wider research community.

However, the GitHub archives provided need some additional work before publication. Only minimal documentation is provided for the GP Jupyter notebooks, and no readme is provided at all for the DengueSim itself. Prior to publication, I'd ask the authors to ensure that these two repositories are documented for maximum accessibility, including providing installation/running instructions and links to related literature and tutorials for beginners to these kinds of modelling techniques.

Overall, I would say that the paper is important for the epidemiological community, where there has been some resistance to the use of complex systems methods, and demonstrating that tools exist to ease the 'curse of dimensionality' problem and provide useful analyses of their outputs will be helpful in driving the community to explore these methods more confidently. Having said that, very similar papers to this exist in other fields (demography, public health, etc.), but given the general lack of interdisciplinary communication regarding methodology I believe this paper is still valuable for its intended readership. Therefore, if the clarifications and additions suggested above are implemented, I would suggest the editors approve this paper for publication.

Reviewer #2: Although I think the methods developed in the article are worth exploring, there are issues with the writing of this manuscript. Every section is too long, which makes the manuscript hard to follow. The results section starts with what is described as a 'brief' description of the results, but is about three pages long. Most of the results section is a mixture of methods and results. Much of the current methods section could be supplementary material. Information on the Gaussian processes should be more prominent.

The article advertises itself as the first paper to use gaussian process emulation to conduct sensitivity analysis. However, Golumbeanu et al [28] is referenced and also uses these methods with Sobol's sensitivity analysis.

Reviewer #3: This paper showcases Gaussian Process (GP) emulation as a way to make sensitivity analysis and systematic exploration tractable for complex, high-dimensional and computationally intensive individual-based models (IBMs): two tasks that would otherwise be practically impossible.

The authors trained GPs as surrogate models on input–output data from the IBM to predict three key outcomes: outbreak probability, peak incidence, and epidemic duration.

While the GPs can efficiently predict IBM outcomes and enable comprehensive sensitivity analysis, their predictive power was limited when applied to real dengue data from Colombia, most likely because the underlying IBM was highly simplified. Still, the approach identified municipalities with consistently higher infectivity estimates that overlapped with known dengue clusters, demonstrating value for capturing broad epidemiological patterns.

In my opinion, the main interest of this work lies in its methodological approach, which is clearly described and serves as a valuable proof-of-concept and a useful reference for applying Gaussian Process emulation to other, more complex IBMs. As the authors note, similar uses of GPs have already been applied to other disease specific IBMs, but their work demonstrates how the method can be applied more broadly to other existing, complex IBMs.

**Have the authors made all data and (if applicable) computational code underlying the findings in their manuscript fully available?**

Reviewer #1: Yes

Reviewer #2: Yes

Reviewer #3: Yes

PLOS authors have the option to publish the peer review history of their article (what does this mean? ). If published, this will include your full peer review and any attached files.

**Do you want your identity to be public for this peer review?** For information about this choice, including consent withdrawal, please see our Privacy Policy .

Reviewer #1: **Yes: ** Dr Eric Silverman

Reviewer #2: **Yes: ** Emma Fairbanks

Reviewer #3: No

**Figure resubmission:**

**Reproducibility:**



---

## [Decision Letter · Decision Letter 1]

18 Dec 2025

Dear Ms. Langmüller,

We are pleased to inform you that your manuscript 'Gaussian Process emulation for exploring complex infectious disease models' has been provisionally accepted for publication in PLOS Computational Biology.

Best regards,

Jennifer A. Flegg

Section Editor

PLOS Computational Biology

Jennifer Flegg

Section Editor

PLOS Computational Biology

Reviewer's Responses to Questions

**Comments to the Authors: 
Please note here if the review is uploaded as an attachment.**

Reviewer #3: I am satisfied with the changes. I do think the paper is now easier to follow.

**Have the authors made all data and (if applicable) computational code underlying the findings in their manuscript fully available?**

Reviewer #3: Yes

PLOS authors have the option to publish the peer review history of their article (what does this mean? ). If published, this will include your full peer review and any attached files.

**Do you want your identity to be public for this peer review?** For information about this choice, including consent withdrawal, please see our Privacy Policy .

Reviewer #3: No

---

## [Editor Report · Acceptance letter]

PCOMPBIOL-D-25-01169R1

Gaussian Process emulation for exploring complex infectious disease models

Dear Dr Langmüller,

I am pleased to inform you that your manuscript has been formally accepted for publication in PLOS Computational Biology. Your manuscript is now with our production department and you will be notified of the publication date in due course.

With kind regards,

Anita Estes
